# Antioxidant, Whitening, Antiwrinkle, and Anti-Inflammatory Effect of *Ajuga spectabilis* Nakai Extract

**DOI:** 10.3390/plants12010079

**Published:** 2022-12-23

**Authors:** Min Sung Lee, Yu Jin Oh, Jae Woo Kim, Kyung Min Han, Da Som Kim, Ji Won Park, Hyeok Mo Kim, Dae Wook Kim, Yeong-Su Kim

**Affiliations:** Division of Industrialization Research, Baekdudaegan National Arboretum, Bonghwa-gun 36209, Republic of Korea

**Keywords:** endemic plant, antioxidant, anti-aging, inflammatory cytokine, *Ajuga spectabilis* Nakai

## Abstract

Since ancient times, plants have been a good source of natural antioxidants. Plants remove active oxygen through antioxidants and contain various active ingredients. These active ingredients of plants are used to alleviate skin aging and chronic diseases. *Ajuga spectabilis* Nakai (AS) is a perennial plant, is endemic to Korea, and has the characteristics of alpine plants. The aim of this study was to assure the possibility of using AS as a functional natural and cosmetic material. For this, we carried out biologically activated material characteristic evaluations about antioxidant, wrinkle reduction, and anti-inflammatory effects using AS extract. To carry out this experiment, we extracted AS extract from AS water extract (AS-W) and AS 70% ethanol extract (AS-E). AS-E showed the highest DPPH activity and tyrosinase inhibitory activity. After, the measurement of metalloprotease (MMP)-1 inhibition effect showed the AS-W and AS-E activation at the concentration of 100 µg/mL. In addition, at the same concentration, from the result of the measurement of the biosynthesis quantity of pro-collagen type-1 we knew that its excellent effect appeared in AS-E (CCD-986sk). The inhibition of NO production in AS-W and AS-E was confirmed in LPS-induced mouse macrophage RAW264.7 cells. On cell viability, it was judged that AS-E had no toxicity because it showed a high cell viability at a high concentration, and it was used for the anti-inflammatory activity. Inhibition of NO production worked only in AS-E; inflammatory cytokine TNF-α and IL-6 were suppressed in a concentration-dependent manner in AS-E. AS is believed to be used as a natural cosmetic material because it has been proven to have antioxidant, whitening, wrinkle-improving, and anti-inflammatory effects. Therefore, the results indicate that AS extract can play an important role as a functional natural material and a cosmetic material for whitening, wrinkle reduction, and anti-inflammatory effect.

## 1. Introduction

As the standard of living and average life expectancy are extended along with economic growth, modern people are making more efforts to manage and maintain their health and beauty. Air pollutants such as fine dust, ultrafine dust, and yellow dust generated by industrial development adsorb to the skin, weaken the skin barrier, cause tissue damage, promote aging, and inflammation, thus reducing stratum corneum function. Therefore, interest in external factors such as skin care is increasing, and the industry for functional materials capable of suppressing skin damage and aging is continuously growing [1]. In addition, reactive oxygen species (ROS) stimulate melanin production [2]. ROS are generated during normal metabolism but excessive production of oxygen due to the imbalance of oxidation-antioxidant action in the body causes oxidative stress. It has been reported that when oxidative stress accumulates continuously, it causes damage to cells, DNA, proteins, and lipids, resulting in diseases such as aging, cancer, and diabetes [3,4].

ROS are frequently generated by environmental stresses such as ultraviolet rays, salt, and high temperatures [5]. Since ROS that can damage plants are generated by such environmental stress, plants may suppress damage caused by ROS through antioxidants, such as polyphenolic compounds, as an antioxidant defense system to counter oxidative stress [6]. Types of antioxidants that have the function of scavenging active oxygen include water-soluble antioxidants such as vitamin C, glutathione, and uric acid; fat-soluble antioxidants tocopherol, carotenoids, and flavonoid/flavonoids; synthetic antioxidants tertbutylhydroxytoluene (BHT), tert-butylhydroxyanisol (BHA); and antioxidant enzymes such as superoxide dismutase (SOD), catalase, glutathione peroxidase and the like [7], which can protect cells from oxidative damage by removing active oxygen intermediates and resist cell apoptosis [8,9].

Melanin is a pigment that determines hair, pupil, and skin color in the human body [10]. It protects the body by absorbing the ultraviolet rays and blocking harmful ultraviolet rays, thus preventing them from penetrating the body [11]. However, excessive accumulation of melanin causes hyperpigmentation, such as spots and freckles, and accelerates skin aging. Melanin synthesis occurs in melanosome found in the melanocytes and it is produced by three enzymes: tyrosinase, tyrosinase related protein-1 (TRP-1), and tyrosinase related protein-2 (TRP-2) using tyrosine as a substrate [12]. Microphthalmia-associated transcription factor (MITF), a known melanogenesis regulator, binds to tyrosinase and regulates the expression of related enzymes [13].

Among the methods for demonstrating the possibility of preventing wrinkles, the degree of biosynthesis of collagen, a component of the dermis using human fibroblasts, and the activity of MMP-1 (Matrix Metallo Proteinase-1), known as an enzyme that degrades collagen, are inhibited to activate antioxidant activity; measurements are recommended [14]. In addition, inflammatory cytokines TNF-a and IL-6 induce an inflammatory response in cells and activate MMPs, which are collagen-degrading enzymes, to degrade collagen and reduce skin elasticity. It can delay the aging process and improve wrinkles [15,16]. Therefore, MMP-1 and pro-collagen synthesis were confirmed using human fibroblast CCD-986sk, and anti-inflammatory activity was evaluated using macrophage RAW264.7, which is responsible for immune and inflammatory responses.

*Ajuga spectabilis* Nakai (AS), (family Lamiaceae, genus *Ajuga*) is a rare perennial herb growing in the southern mountains of Korea and has wide, ellipsoidal leaves and deep violet flowers [17,18]. AS is a flowering tree used for landscaping or horticulture because it has thin leaves, purple flowers, and a high ornamental value. According to a previous study, Luteolin 5-glucoside, a compound isolated from AS, suppressed ROS generation and demonstrated a protective effect against oxidative stress-induced cytotoxicity [19]. According to the research result of Chung et al. [20], iridoid glycoside was isolated from the extract of AS, and the isolated compound was named ‘Jaranidoside’. However, since there is no research based on the activity and cells of the whole plant of AS, this study aims to inform the efficacy of Korean endemic plant and provide basic data. As there are scarce physicochemical studies supporting AS’s protective effects, this study confirms AS’s antioxidant, whitening, antiwrinkling, and anti-inflammatory activities of the water (AS-W) and ethanol extract (AS-E). 

In this way, despite the fact that the components of AS are recognized as materials with physiological usefulness in the existing reported studies, studies as cosmetic materials or functional materials are insufficient. Although the clear mechanisms of antioxidation, anti-aging, and anti-inflammatory effects of AS extract have not yet been identified, it is considered worthy of attention. 

Therefore, in this study, we intend to utilize Korean indigenous plants as basic data for the development of natural cosmetic materials through research on various physiological activities such as antioxidant effect, Tyrosinase and MMP-1 inhibitory activity, procollagen synthesis, and NO production.

## 2. Results

### 2.1. DPPH Radical Scavenging Activity

The radical scavenging activity of the AS-W and AS-E extracts was measured using DPPH. When 0.1 mM DPPH solution was added to 50 μg/mL of AS-W and AS-E, the radical scavenging activity of AS-W and AS-E was 35.8% and 57.5%. The activity was lower than the positive control L-ascorbic acid that had an activity of 89.7%. Results indicate that AS-E had higher antioxidant activity than AS-W (Figure 1).

### 2.2. Tyrosinase Inhibitory Activity 

To determine whether an extract effectively inhibits melanin synthesis, the measurement of mushroom-derived tyrosinase inhibitory activity is known to be a useful method. AS-W extracts (50 μg/mL) reported a tyrosinase inhibitory activity of 31.4% and AS-E extracts (50 μg/mL) reported that of 43.5% (Figure 2). This showed lower activity than Arbutin, a positive control tyrosinase inhibitory activity. Since AS-E has higher tyrosinase inhibitory activity than AS-W, it is believed to have a whitening effect.

### 2.3. Cell Viability and MMP-1 Expression Inhibitory Activity (CCD-986sK)

To evaluate the effect of AS extract on CCD-986sK cell viability, the cells were treated with 200 μg/mL and 100 μg/mL concentrations of AS-W and AS-E. Results indicate that AS-W and AS-E 100 μg/mL had a positive effect on the cell viability. To determine if the expression of the collagen fiber’s enzyme MMP-1 was regulated by AS extracts treatment (Figure 3), fibroblast and CCD-986sk were treated with TNF-α. The TNF-α treatment increased the expression of MMP-1 compared to the control group, but its expression reduced significantly upon AS-W or AS-E treatment. AS-W extracts (100 μg/mL) reported an MMP-1 production of 71.8% and AS-E extracts (100 μg/mL) reported that of 58.8% (Figure 3). This suggests that AS-E has a better inhibitory ability of MMP-1 than AS-W. 

### 2.4. Measurement of Pro-Collagen Synthesis in Fibroblasts (CCD-986sK)

To evaluate the effect of AS extract on CCD-986sK pro-collagen synthesis. The effect on collagen biosynthesis was investigated for 100 μg/mL concentration (Figure 4) of AS extract by comparing it with the control. Results indicate that collagen biosynthesis increased by 108.8% and 135.6% after AS-W and AS-E treatment, respectively. This is expected to contribute to skin elasticity and antiwrinkle effects.

### 2.5. Cell Viability and NO Production of Macrophages RAW264.7

Cytotoxicity was evaluated in the AS extract treated RAW264.7 cells using MTT assay (Figure 5). Results suggest that AS-W was toxic and reduced cell viability in a concentration-dependent manner. In addition, AS-E was judged to be non-toxic because the cell viability was also high as the concentration of the extract increased. In addition, the effect on NO production, a chronic inflammation mediator, was also investigated by administering AS extract treatment. Results showed that LPS alone treatment significantly increased NO production, AS-W did not affect NO production, and AS-E inhibited the NO generation significantly.

### 2.6. Inhibition of Inflammatory Cytokine (IL-6, TNF-α) 

The effect of AS extract on the inflammatory cytokines showed that AS-E suppressed the NO production. LPS is known to produce an inflammatory response; a co-treatment of AS-E was administered to evaluate inflammation inhibitory potential. Since AS-E showed higher cell viability at higher concentrations than AS-W, the inflammatory cytokine inhibition was evaluated only in AS-E. AS-E was administered at 12.5, 25, or 50 μg/mL, and results suggest IL-6 and TNF-α was inhibited significantly from 12.5–50 μg/mL concentration (Figure 6).

## 3. Discussions

Recently, new factors such as fine dust and COVID-19, along with intrinsic and extrinsic factors such as ultraviolet rays and stress, appear to accelerate aging, thus engendering an interest in anti-aging research [21]. As the average lifespan of humans is increasing, research on antioxidants and natural products that improve wrinkles, skin health, and skin whitening is increasing, the functional cosmetics industry related to antioxidants that inhibit aging is also on the rise [22]. This study explored the antioxidant, whitening, antiwrinkle, and anti-inflammatory properties of the (AS extract of) Korean plant *Ajuga spectabilis* Nakai. Due to the side effects of synthetic antioxidants, products having a natural origin are actively made. As interest in natural products has increased, the use of natural plant materials in functional cosmetics is also increasing [23,24,25].

DPPH is a compound containing free radicals; it is used to measure antioxidant activity in vitro, as it binds to a hydrogen group and changes color to indicate the quenching of free radicals [26]. DPPH radical scavenging activity was used to examine the antioxidant potential of the AS extract. As a result of measurement, both AS extracts AS-W and AS-E showed activity as high as AA, the representative antioxidant. In a study by Ji et al. [27], when the DPPH activity was evaluated in the water and 70% ethanol extract of *Agastache rugosa* of Lamiaceae family, the results indicated that the antioxidant activity of the water extract was 2.5 times higher than that of the ethanol extract.

The tyrosinase enzyme acts at the initial determining step of melanin synthesis and has hydroxylase activity, thus converting tyrosine to 3,4-dihydroxyphenyl alanine (DOPA), and DOPA oxidase oxidizes DOPA to DOPA quinine [28]. Tyrosinase acts as a rate-limiting enzyme in the biosynthesis of the melanin polymer and is of great significance in the early-stage development of whitening materials [29]. In this study, the tyrosinase inhibitory activity of the AS extract was measured using L-DOPA. To reduce the melanin production, either the synthesis of tyrosinase or its activity can be inhibited [30]. Results suggested that both AS-W and AS-E extracts had tyrosinase inhibitory potential when compared with the control arbutin. As a result of Lee’s [31] study, when comparing water and ethanol extracts from the leaves of *Angelica gigas* Nakai, the tyrosinase inhibitory activity was higher in the 70% ethanol extract. As a result, similar to this study, it is believed that plants containing phenolic compounds generally tend to show antioxidant power and inhibit melanin biosynthesis by reducing substances that generate tyrosinase, an oxidizing enzyme [32]. This indicates that AS can impact the melanin production mechanism and is effective as a natural whitening material.

MMP-1 is known to promote wrinkle formation by decomposing collagen type 1, and it is a proteolytic enzyme that specifically acts on collagen to protect collagen and maintain elasticity, thereby preventing wrinkles [33]. Among human skin constituent cells, dermal fibroblasts are responsible for the expression of fibrous proteins, collagen, and elastin, and both natural aging and photoaging cause wrinkles to form and collagen fibers in the dermis to decrease [34]. Therefore, human fibroblast CCD-986sk cells were used in the experiment. This study showed the inhibitory effect of AS-W and AS-E extracts on MMP-1, i.e., they prevent degradation of collagen among the components constituting the connective tissue of skin cells in fibroblasts. In a study by Hong et al. [35], similar to AS extract, *Prunella vulgaris* extract showed similar results by inhibiting the MMP-1 activity. In skin, the synthesis of type I collagen and the activity of MMP-1, a decomposing enzyme, thereof, are balanced. However, in the skin tissue, due to external and internal causes, the synthesis of type I and III collagen lowers, and MMP-1 activity increases while aging [36]. Therefore, the promotion of collagen production is considered to be a way to prevent aging [37]. In the cytotoxicity experiments, the AS-W and AS-E extracts maintained the cell viability close to 100% at the concentration of 100 ug/mL, so the concentration selected was set to 100 ug/mL. As a result, it was confirmed that the AS-W and AS-E extracts were effective in improving wrinkles by keeping the cells viable. 

To investigate the effect of AS extract on collagen synthesis, procollagen synthesis was assessed. Results confirm that the pro-collagen synthesis increased in response to 100 μg/mL AS extract. In particular, it was observed that AS-E significantly increased the synthesis. AS-E and its extract were proven to be effective natural ingredients for improving skin aging by inhibiting collagen degradation and promoting collagen biosynthesis by regulating the activity of MMP-1 by ultraviolet rays. In a study by Um [38], it is reported that the stem and root extracts of *Achyranthes japonica* have a positive effect on collagen synthesis compared to the control group. This study also showed a significant improvement in collagen synthesis, so it can be considered that AS is effective in wrinkle improvement. 

Macrophages play an important role in regulating immune function and inflammatory response. Excessive production of NO causes inflammatory diseases and oxidative damage to cells and tissues, leading to genetic mutations and nerve damage. Thus, inhibiting NO production is a known method that can inhibit the onset of inflammation-related diseases. Upon induction of inflammation, macrophages defend against cytokines, which are active protein substances, and IL-6 and TNF-α are known as representative inflammatory cytokines [39]. TNF-α causes fatal toxicity when overexpressed, causing sepsis and skin inflammation [40]. IL-6 affects chronic inflammation, fat metabolism, and insulin resistance, and the expression of IL-6 induces other inflammatory cytokines such as TNF-a and IL-1 [41,42]. In this study, the ability of the AS extract to inhibit NO production and inhibit inflammatory cytokines was confirmed. When the inflammatory substance LPS was induced in RAW264.7, a mouse macrophage, NO production was inhibited in a concentration-dependent manner in both the AS-W and AS-E extracts. Between the two, it was confirmed that the AS-E significantly inhibited NO production. As a result of the previous cytotoxicity test, toxicity was only not observed in AS-E at concentrations of 12.5–50 μg/mL, which is considered to confirm that the NO production-inhibitory effect of AS-E extract is not due to the decrease in cell population due to toxicity. A study by Oztuurk and Ozbek [43] also reported that the ethanol extract of *Eugenia caryophyllata* showed a higher inhibition of NO production than the water extract. Since difference in the content of physiologically active substances contained in natural products depends on the extraction solvent [44], a large amount of polyphenols are extracted during ethanol extraction compared to water extraction, which is consistent with this study. Hence, the content of active ingredients in the ethanol extract is higher. In a study by Kim et al. [45], when the expression of TNF-a and IL-6 was confirmed according to the extraction method of *Portulaca oleracea*, it was confirmed that the expression amount decreased in ethanol extract. This is a result consistent with previous studies, and it is considered that natural products are affected by the production-inhibitory activity of NO and anti-inflammatory cytokines depending on the extraction solvent.

In conclusion, antioxidant, whitening, antiwrinkle, and anti-inflammatory effects were confirmed in AS extract. Therefore, in future, additional research and data acquisition will be required to understand the mechanism of action for the anti-aging effect during AS treatment. Hence, it can be believed that AS can provide basic information that can be used as a natural cosmetic and industrial material.

## 4. Materials and Methods

### 4.1. Chemical Reagents

The 2,2-diphenyl-1-picrylhydrazyl (DPPH) was obtained from Sigma Aldrich (St. Louis, MO, USA). Mushroom tyrosinase (Sigma Chemical Co., USA). Matrix metalloproteinase-1 ELISA kit (Abcam, Cambridge, MA, USA), procollagen type-C peptide (PIP) EIA kit (Takara, Shiga, Japan), 3-(4,5-dimethylthiazol-2-yl)-2,5-diphenyltetrazolium bromide (MTT), phosphate-buffered saline (PBS), and MMP-1 were purchased from Invitrogen (Carlsbad, CA, USA). Dulbecco’s modified Eagle’s medium (DMEM), Fetal bovine serum (FBS) and Ham’s F-12 nutrient mixture, L-Glutamine, and 4-(2-hydroxyethyl)-1-piperazineethanesulfonic acid buffer (DMEM/F-12) were purchased from Lonza (Walkersville, MD, USA). LPS was purchased from Sigma-Aldrich (St. Louis, MO, USA). TNF-a and IL-6 were from ELISA Kit (BioLegend, San Diego, CA, USA), and the RAW264.7 mouse machrophage cells were from the Korean Cell Line Bank (Seoul, Republic of Korea). CCD-986sk human fibroblast cells were from American Culture Collection (Manassas, VA, USA).

### 4.2. Sample Preparation

The AS sample was collected from Bonghwa-gun, Gyeongsangbuk-do, Republic of Korea, in July 2017. After lyophilization, 400 mL hot water and 400 mL 70% ethanol were added to 20 g of the powdered AS. The solution was AS-W heated at 80–100 °C for 3 h and AS-E was extracted in a shaking incubator for 48 h. After extraction, the sample was filtered (No. 2, Whatman Co., Maidstone, Kent, UK), concentrated in a vacuum evaporator in a bath at below 40 °C, and lyophilized. The sample extract was then dissolved in dry water and dimethyl sulfoxide (DMSO) and used in subsequent experiments.

### 4.3. Cell Culture

Human-derived fibroblasts cells (CCD-986sK), and mouse macrophages (RAW264.7) were purchased from the Korean Cell Line Bank (Seoul, Republic of Korea). Cells were cultured in a DMEM/F-12 medium containing penicillin, 10% fetal bovine serum, and 100 µg/mL streptomycin, at 5% CO_2_. After reaching 80% confluency, the cells were passaged using trypsin-ethylenediaminetetraacetic acid solution.

### 4.4. DPPH Radical Scavenging Activity

The DPPH radical scavenging ability was measured by modifying the Blois [46] method. Briefly, 90 μL DPPH solution and 10 μL of the AS-W and AS-E extracts were mixed and then incubated at 37 °C for 30 min, and absorbance was measured at 515 nm (UV/Visible spectrophotometer, Perkin Elmer, Waltham, MA, USA). The experiment was replicated in triplicates and expressed as percentages (%).

DPPH radical scavenging activity (%) = (1 − Absorbance of the sample added group/Absorbance of the non-added group) × 100.

### 4.5. Tyrosinase Inhibitory Activity

Tyrosinase inhibitory activity was measured using the method of Yagi et al. [47]. Here, 100 μL of 0.2 M phosphate buffer (pH 6.5), 50 μL of 2 mM L-tyrosine, 20 μL of sample, and 10 μL of 200 unit/mL tyrosinase were sequentially mixed and reacted at 37 °C for 30 min, and the absorbance was measured at 470 nm.
Tyrosinase activity (%) = (1 − sample group/control group) × 100

### 4.6. MMP-1 Expression Inhibition Activity

The MMP-1 synthesis assay was tested using Gross et al.’s [48] method. CCD-986sk cells were seeded in a 96-well plate at a density of 1 × 10^5^ cells/well and incubated for 24 h. After incubation, the medium was replaced with 1 mL of 10% PBS and irradiated with UV light. To measure MMP-1 activity, 10 ng/mL of TNF-α was added to each well, followed by the addition of AS-W (100 µg/mL) and AS-E extracts (100 µg/mL). After 48 h of incubation, the activity was measured using the MMP-1 Activity Assay Kit (Biotrack; Amersham Bioscience, Piscataway, NJ, USA).

### 4.7. Procollagen Synthesis Activity

Procollagen synthesis assay was assessed using the method of Krupsky et al. [49]. CCD-986sk cells were seeded in a 96-well plate at a density of 1 × 10^4^ cells/well and cultured for 24 h. Cells were washed with 1 × PBS twice and a serum-free medium was added. After 24 h, AS-W and AS-E extracts were added and incubated for 48 h. The quantity of procollagen synthesized was measured by adding supernatant to procollagen Type-I C-Peptide EIA kit (Takara-Bio Inc., Shiaga, Japan).

### 4.8. Measurement of NO Production Inhibition

NO production was measured using the method described by Namkoong et al. [50], with some modifications. Mouse macrophage RAW264.7 cells were seeded in a 12-well plate and incubated for 24 h at 37 °C. Subsequently, the cells were treated with 12.5, 25, and 50 µg/mL of AS-W and AS-E and incubated for 6 h at 37 °C. After incubation, the cells were treated with 1 µg/mL LPS and cultured for 18 h at 37 °C. NO production was measured using the Griess reagent (Sigma-Aldrich Co., St. Louis, MO, USA). Absorbance was measured at 540 nm using a microplate reader (PerkinElmer, Waltham, MA, USA).

### 4.9. Measurement of Cell Viability

Cell viability was measured using the MTT assay in RAW264.7 cells and CCD-986sk. The cells were seeded at a density of 1 × 10^6^ cells/well in 12-well culture plates and treated with AS-W and AS-E for 12 h. Subsequently, the cells were incubated with 200 µL of MTT solution (1 mg/mL) for 2 h. After incubation, the MTT solution was removed and the resulting crystals were dissolved in DMSO. The absorbance was measured at 570 nm (Perkin Elmer, Waltham, MA, USA).

### 4.10. Inflammation Cytokine Inhibitory Measurement

The mouse macrophage RAW264.7 cells were prepared the same as for the NO measurement, and the supernatant was collected from each well after giving extract treatment. TNF-α and IL-6 were quantified as per the manufacturer’s guidelines (Mouse IL-6 ELISA kit or a Mouse TNF-α ELISA kit purchased from ENZO bioscience, Farmingdale, NY, USA). The absorbance was measured with an ELISA reader, and TNF-α and IL-6 were quantified using a standard curve.

### 4.11. Statistical Analysis

All data are presented as mean ± standard deviation (SD). The differences between treatments were determined with a Student’s *t*-test, and *p* < 0.05 was considered statistically significant.

## 5. Conclusions

In this study, a hot water extract (AS-E) and an ethanol extract (AS-E) were finally obtained using purified water and 70% ethanol. The findings obtained from this study demonstrate that AS is a potential source of phenols and flavonoid, phytochemicals, and various compounds. The current study showed that AS exerts antioxidant effects through DPPH and tyrosinase inhibition activity, suppresses the expression of MMP-1 in CCD-986sk cells stimulated with TNF-α, and increases pro-collagen synthesis, resulting in anti-aging activity. AS inhibits the production of NO in a concentration-dependent manner and inhibits inflammatory cytokines (IL-6, TNF-α) in LPS-stimulated mouse macrophage RAW264.7 cells, suggesting that AS has potential for the treatment and prevention of inflammatory skin diseases. We aim to use it as an opportunity to produce AS-derived products, realize their value addition by applying it to various industrial fields, and to provide information about Korea’s valuable and unique natural resources. These findings suggest that AS extracts are a potent agent for the development of natural cosmetics and functional foods with antioxidants, anti-aging, and anti-inflammatory properties. 

## Figures and Tables

**Figure 1 plants-12-00079-f001:**
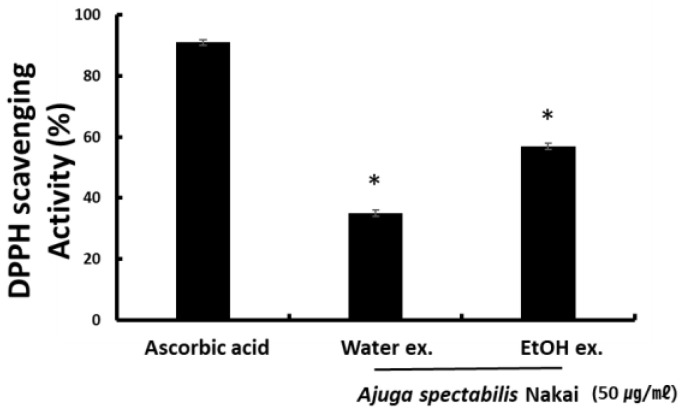
DPPH radical scavenging activity showing antioxidant effects of AS-W, AS-E. Data are presented as mean ± SD. * *p* < 0.05 compared with control.

**Figure 2 plants-12-00079-f002:**
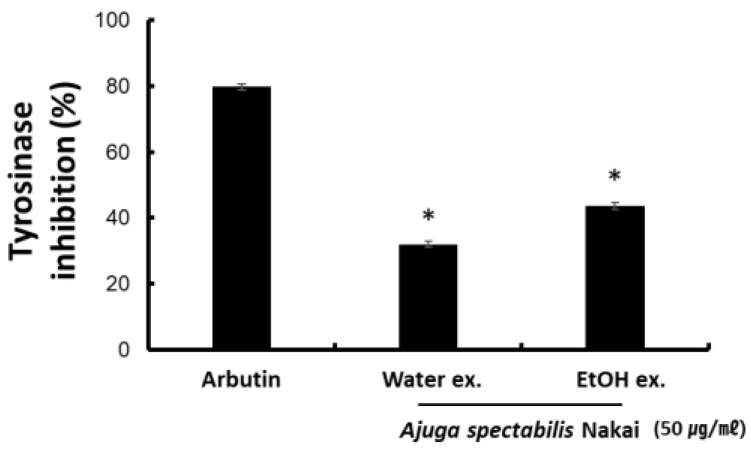
Effect of AS-W, AS-E on the inhibition of tyrosinase production of as measured AS-W, AS-E. Data are presented as mean ± SD. * *p* < 0.05 compared with control.

**Figure 3 plants-12-00079-f003:**
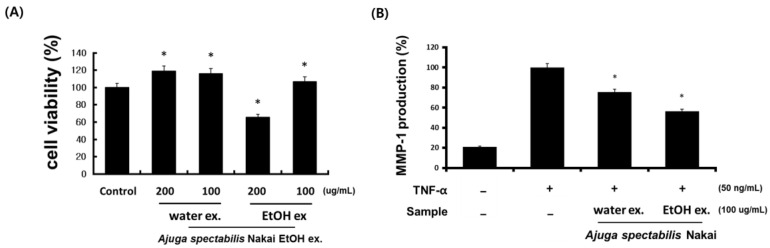
Effect of AS-W, AS-E on cell viability (**A**) and MMP-1 inhibition (**B**) in CCD-986sk fibroblasts. The cells were treated with various concentration of AS-W, AS-E for 48 h. * *p* < 0.05 compared with control.

**Figure 4 plants-12-00079-f004:**
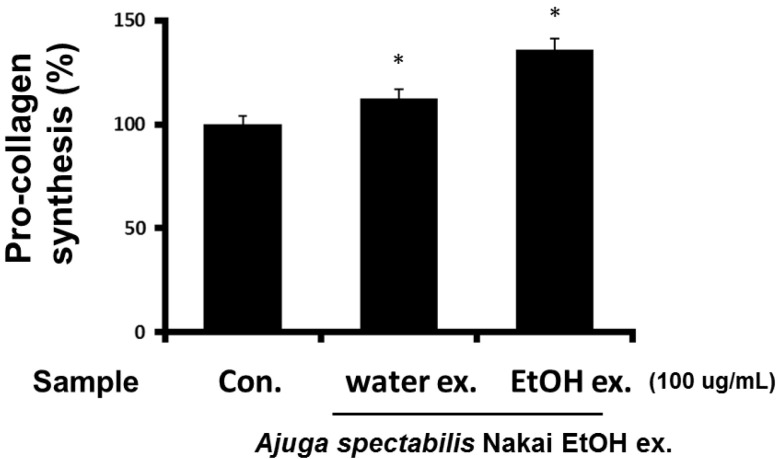
Effect of AS-W, AS-E on procollagen synthesis in CCD-986sk fibroblasts. The cells were incubated for 24 h in DMEM medium containing 10% FBS, followed by incubation with various concentrations of AS-W and AS-E for 48 h. * *p* < 0.05 compared with control.

**Figure 5 plants-12-00079-f005:**
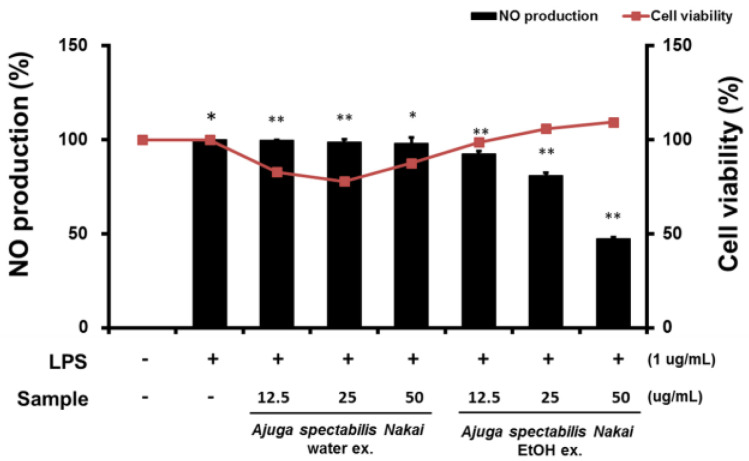
Determining NO production and cell viability in RAW264.7 cells. Cells were pre-treated with AS-W, AS-E for 6 h and then co-treated with LPS (1 μg/mL) for 18 h. * *p* < 0.05 compared to the cells without the treatment, and ** *p* < 0.05 compared to the cells treated with LPS alone.

**Figure 6 plants-12-00079-f006:**
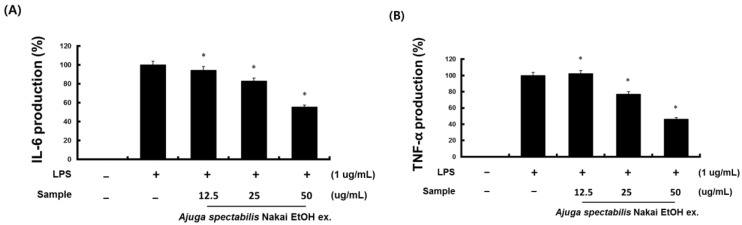
Inhibitory effect of AS-E on the production of IL-6 (**A**), TNF-α (**B**) in RAW264.7 cell. RAW264.7 cells were pre-treated with AS-E for 6 h and then co-treated with LPS (1 μg/mL) for 24 h. *p* < 0.05 compared to the cells without the treatment, and * *p* < 0.05 compared to the cells treated with LPS alone.

## Data Availability

Not applicable.

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
