# Peer review of "Antioxidant, Whitening, Antiwrinkle, and Anti-Inflammatory Effect of Ajuga spectabilis Nakai Extract"

_plants, 2022, doi:10.3390/plants12010079_

Round 1
Reviewer 1 Report
Dear authors,
For starters I want to congratulate you for the article. The study is interesting, but I think there are numerous issues that must be clarified and corrected before this manuscript is published.
Abstract: Even with the clearly stated aim of the work, the abstract is very general.
Introduction:
- Lines 41-43: Although it's true that ROS can be formed during regular metabolism, it's equally important to note that exogenous factors can also cause ROS generation. This section of the external factors that contribute to the formation of ROS should be presented.
- Lines 46-48: Your intended message is not clear. Please rephrase and be more specific about: the relationship between ROS and the inhibitory activity of the enzymes that are being discussed; the types of antioxidants (enzymatic or non-enzymatic) that can inactivate ROS; and the specific antioxidants that are important for the inhibitory action.
- The introduction presents a single physiologically active chemical related to the plant under research. It is important to provide more information regarding the phytochemical composition of the plant. Is this a medicinal plant? Is it a plant that is used in traditional medicine? What applications does this herb have?
- In the study, three different types of cell cultures were chosen, which were tested in parallel, in each case other biochemical parameters were analyzed. In the introduction, these parameters and their connection with the purpose of the article should be specified. Also, what is the importance of these parameters and what were the reasons why this distribution was chosen in different cell cultures.
- I believe that this chapter presents information in a way that is far too generic; in order to understand the state of the art in this field and the objectives that this study aims to achieve, more details are required.
Materials and Methods:
- The design of this study seems disorganized and incomplete. The extracts were tested on various cell cultures. The authors do not specify the doses administered in each case. Certain very vague specifications appear (e.g. line 244 - AS-H (100 μg/mL) and AS-E extracts (100 μg/mL). What does this concentration mean? What compound does this concentration refer to? The manuscript does not present any characterization of extracts from a qualitative and quantitative point of view.
- Lines 230 – 238 – It's unclear. Which one of the cell lines was the subject of this analysis? All of them or just some of them? The only line that is mentioned in the results is the B16F10 mouse line. It should be mentioned in this chapter as well.
- Lines 229 – Tyrosinase inhibition activity must be removed (I think it's a writing error).
- Lines 240 – 246 - Is MMP1 expressed only in human fibroblasts CCD cells? Why did you decide to study the effects of extract administration on MMP1 just on this specific cell line?
- Lines 255 – 261: See the previous observation.
- Lines 263 – 267: All 3 of the investigated cell lines must undergo cell viability testing. The protocol calls for the determination of this parameter in the situation when we test cell extracts on cultures in vitro to see if there was a toxicity associated with different extract concentrations.
- Lines 269 – 273: What cells?
Results:
- Line 99: Figure 4?
- Line 110: Figure 5: It is not very clear how cell viability varies and no statistical data can be seen. Please modify the chart.
- Line 122: Figure 6?
- Line 123: The text states that both extracts AS-E and AS-H were tested. This is not evident from the presented graph.
Discussions:
- In some cases, the discussions are too general, without referring to other studies.
Author Response
To the editor
thank you very much for investing time and effort in reviewing my manuscript, and it is a great honor to be able to submit a research paper to a good journal called ‘plants’. I tried my best to answer, but my english grammar or expression may be lacking. I hope it didn’t cause you any inconvenience. We ask for your understanding in advance. Thank you.
Reviewre 1
Dear authors,
For starters I want to congratulate you for the article. The study is interesting, but I think there are numerous issues that must be clarified and corrected before this manuscript is published.
Abstract: Even with the clearly stated aim of the work, the abstract is very general.
Introduction:
- Lines 41-43: Although it's true that ROS can be formed during regular metabolism, it's equally important to note that exogenous factors can also cause ROS generation. This section of the external factors that contribute to the formation of ROS should be presented.
Answer) Thank you for your comment .mentioned in Lines 51-56.
- Lines 46-48: Your intended message is not clear. Please rephrase and be more specific about: the relationship between ROS and the inhibitory activity of the enzymes that are being discussed; the types of antioxidants (enzymatic or non-enzymatic) that can inactivate ROS; and the specific antioxidants that are important for the inhibitory action.
Answer) mentioned in Lines 56-59.
- The introduction presents a single physiologically active chemical related to the plant under research. It is important to provide more information regarding the phytochemical composition of the plant. Is this a medicinal plant? Is it a plant that is used in traditional medicine? What applications does this herb have?
Answer) My research materials are very incomplete. This is a Korean specialty plant. mentioned in Lines 67-79
- In the study, three different types of cell cultures were chosen, which were tested in parallel, in each case other biochemical parameters were analyzed. In the introduction, these parameters and their connection with the purpose of the article should be specified. Also, what is the importance of these parameters and what were the reasons why this distribution was chosen in different cell cultures.
Answer) The cells used are mentioned in the discussion.
- I believe that this chapter presents information in a way that is far too generic; in order to understand the state of the art in this field and the objectives that this study aims to achieve, more details are required.
Answer) Based on this research, we plan to use it as basic data, and to proceed with cell-based mechanism research, analysis, and animal experiments in the future.
Materials and Methods:
- The design of this study seems disorganized and incomplete. The extracts were tested on various cell cultures. The authors do not specify the doses administered in each case. Certain very vague specifications appear (e.g. line 244 - AS-H (100 μg/mL) and AS-E extracts (100 μg/mL). What does this concentration mean? What compound does this concentration refer to? The manuscript does not present any characterization of extracts from a qualitative and quantitative point of view.
Answer) Corrected by not mentioning the sample concentrations used in the DPPH and tyrosinase experiments. AS-W(100 μg/mL) and AS-E(100 μg/mL) of the MMP-1 method mean the concentration of the extract.
- Lines 230 – 238 – It's unclear. Which one of the cell lines was the subject of this analysis? All of them or just some of them? The only line that is mentioned in the results is the B16F10 mouse line. It should be mentioned in this chapter as well.
Answer) There was confusion in the experimental method. I used the mushroom derived tyrosinase inhibitory method. Corrected.
- Lines 229 – Tyrosinase inhibition activity must be removed (I think it's a writing error).
Answer) Modified method.
- Lines 240 – 246 - Is MMP1 expressed only in human fibroblasts CCD cells? Why did you decide to study the effects of extract administration on MMP1 just on this specific cell line?
Answer) Experiments were conducted using dermal layer cells, which are one of the cells constituting human skin and are related to skin aging. mentioned in Lines 185-189.
- Lines 255 – 261: See the previous observation.
- Lines 263 – 267: All 3 of the investigated cell lines must undergo cell viability testing. The protocol calls for the determination of this parameter in the situation when we test cell extracts on cultures in vitro to see if there was a toxicity associated with different extract concentrations.
Answer) Added cell viability activity data and confirmed no toxicity.
- Lines 269 – 273: What cells?
Answer) mentioned and corrected.
Results:
- Line 99: Figure 4?
Answer) entered incorrectly. Modified.
- Line 110: Figure 5: It is not very clear how cell viability varies and no statistical data can be seen. Please modify the chart.
Answer) mentioned in Lines 126-127.
- Line 122: Figure 6?
Answer) entered incorrectly. Modified.
- Line 123: The text states that both extracts AS-E and AS-H were tested. This is not evident from the presented graph.
Answer) Modified. When evaluating toxicity, only AS-E with high cell viability was applied.
Discussions:
- In some cases, the discussions are too general, without referring to other studies.
Answer) Unreferenced parts are additionally mentioned.
Reviewer 2 Report
Abstract: It's short but not synthetic. Many elements were missing that would summarize the whole experiment. In its present form, it is an excerpt from individual chapters. Authors must write a decent abstract with an introduction that explains the background to the problem, the methods they used, the results, and conclusions.
Authors should improve keywords. They are now a repeat of the title.
The introduction is quite good at the moment, but there is no indication of how this article differs from many others. Please indicate the novelty of your research, which has been described. It is not known whether the results obtained are significantly different or not. The authors do not attempt to interpret their data. These are generally only statements about the results obtained.
Discussion: Based on 21 reports, well conducted.
Materials and Methods: the whole chapter needs to be corrected. First, there are no references to the literature. Did the authors of the manuscript develop all analytical procedures themselves? I think not. Please indicate literature references in all subsections. The procedure should also be described in detail, including modifications, in the following subsections:
4.3. cell culture; 4.4. DPPH radical scavenging activity; 4.5. tyrosinase activity inhibitors; 4.6. MMP-1 expression inhibition activity; 4.7. Procollagen synthesis activity; 4.8. Measurement of NO production inhibition; 4.9. Measurement of cell viability; 4.10. Inflammation cytokine inhibitors measurement.
Conclusion: Authors are asked to formulate 3-4 conclusions that result from the conducted experiment. Conclusions that would be an indication for practice.
Author Response
Q1. Abstract: It's short but not synthetic. Many elements were missing that would summarize the whole experiment. In its present form, it is an excerpt from individual chapters. Authors must write a decent abstract with an introduction that explains the background to the problem, the methods they used, the results, and conclusions.
Answer) Thank you for your comment. I’ve added some of your suggestions (Line 19-21). I’ve provided a more detailed method in Methods and materials, and I just wanted to briefly summarize it in the abstracts. Thank you.
:
Q2. Authors should improve keywords. They are now a repeat of the title.
Answer) Thank you for your comment. Edited keywords.
Q3. The introduction is quite good at the moment, but there is no indication of how this article differs from many others. Please indicate the novelty of your research, which has been described. It is not known whether the results obtained are significantly different or not. The authors do not attempt to interpret their data. These are generally only statements about the results obtained.
Answer) Thank you for your concern. My research plant material is a situation in which there are few prior studies. Therefore, I think that I can provide basic data for starting this study. Additional details are mentioned in detail (Line 60-64).
Q4. Materials and Methods: the whole chapter needs to be corrected. First, there are no references to the literature. Did the authors of the manuscript develop all analytical procedures themselves? I think not. Please indicate literature references in all subsections. The procedure should also be described in detail, including modifications, in the following subsections:
(4.3. cell culture; 4.4. DPPH radical scavenging activity; 4.5. tyrosinase activity inhibitors; 4.6. MMP-1 expression inhibition activity; 4.7. Procollagen synthesis activity; 4.8. Measurement of NO production inhibition; 4.9. Measurement of cell viability; 4.10. Inflammation cytokine inhibitors measurement.)
Answer) Thank you for your comment. I referenced the methods of other researchers and experimented with variations. The material reference I sent you earlier was classified as ‘further reading’ but there was no reference number. Corrected. Thank you.
Q5. Conclusion: Authors are asked to formulate 3-4 conclusions that result from the conducted experiment. Conclusions that would be an indication for practice.
Answer) Thank you for your comment. I have edited your suggestion.
Reviewer 3 Report
The article needs revision.
you should uniform the terms: anti-oxidant - antioxidant; anti-wrinkle - antiwrinkle
Abstract line 10 and 11: is a perennial plants... is a perennial plant - it is not plural; as well endemic plants, singular.
Keywords: CCD-986sk - this is not mentioned in the abstract.
The article needs English language editing.
Line 52: Labitae??? This is ancient term now, the plant belongs to Lamiaceae family.
You need to present chemical composition of the plant extract. The journal Plants is with high impact factor.
Although, biological assays are good, we need to see chemical composition of the extracts used, to be able to link bioactivities with compounds. You wrote that one flavonoid was detected, but you need to present in this manuscript chemical characterization.
Author Response
Q1. The article needs revision. you should uniform the terms: anti-oxidant - antioxidant; anti-wrinkle – antiwrinkle. Abstract line 10 and 11: is a perennial plants... is a perennial plant - it is not plural; as well endemic plants, singular. Keywords: CCD-986sk - this is not mentioned in the abstract.The article needs English language editing. Line 52: Labitae??? This is ancient term now, the plant belongs to Lamiaceae family.
Answer) I have edited your suggestion.
Q2. Keywords: CCD-986sk - this is not mentioned in the abstract.
Answer) Modified
Q3. The article needs English language editing.
Answer) Modified
Q4. Line 52: Labitae??? This is ancient term now, the plant belongs to Lamiaceae family.
Answer) Modified
Q2. You needd to present chemical composition of the plant extract. The journal Plants is with high impact factor. Although, biological assays are good, we need to see chemical composition of the extracts used, to be able to link bioactivities with compounds. You wrote that one flavonoid was detected, but you need to present in this manuscript chemical characterization.
Answer) Thank you for your concern. My research plant material is a situation in which there are few prior studies. Therefore, I think that I can provide basic data for starting this study. Therefore, it was mentioned that we plan to conduct additional research on related mechanisms and components in the future. Regarding previous research, it is additionally mentioned in the introduction and mentioned in the discussion.
Round 2
Reviewer 1 Report
Abstract: Even with the clearly stated aim of the work, the abstract is very general.
The abstract was not changed or improved in any way by the authors.
Introduction: In the study, three different types of cell cultures were chosen, which were tested in parallel, in each case other biochemical parameters were analyzed. In the introduction, these parameters and their connection with the purpose of the article should be specified. Also, what is the importance of these parameters and what were the reasons why this distribution was chosen in different cell cultures.
Authors answer: The cells used are mentioned in the discussion.
The authors did not make any changes. The introduction of these requirements in the discussion chapter is not relevant to a reader. These data must be mentioned initially, in the objectives of the study.
Introduction: I believe that this chapter presents information in a way that is far too generic; in order to understand the state of the art in this field and the objectives that this study aims to achieve, more details are required.
Authors answer: Based on this research, we plan to use it as basic data, and to proceed with cell-based mechanism research, analysis, and animal experiments in the future.
The authors did not make any changes. These data must be mentioned initially, in the objectives of the study.
Materials and Methods: The design of this study seems disorganized and incomplete.
The authors did not make any changes.
Author Response
I am very sorry for the late reply. Thanks to you, I was able to look through the journal once again. Thank you very much for taking the time to rate the journal.
1. Abstract: Even with the clearly stated aim of the work, the abstract is very general.
The abstract was not changed or improved in any way by the authors.
Answer: Thank you your point made me take a closer look at my journal. Abstract has been changed. I wrote a little more specifically and only mentioned the key points about my research. However, if it is still a problem that you wrote 'general', please tell me what part is general in detail.
2. Introduction: In the study, three different types of cell cultures were chosen, which were tested in parallel, in each case other biochemical parameters were analyzed. In the introduction, these parameters and their connection with the purpose of the article should be specified. Also, what is the importance of these parameters and what were the reasons why this distribution was chosen in different cell cultures.
Answer: thank you. The contents of Introduction have been changed. The reason why I used 3 different cells was mentioned in the introduction. (Line 63-72)
3. Introduction: I believe that this chapter presents information in a way that is far too generic; in order to understand the state of the art in this field and the objectives that this study aims to achieve, more details are required.
Answer: thank you The content has been changed based on your comments. Nevertheless, if you think it is too "genetic", please let me know how to make it more specific. Please. There is no research on the physiological activity of this plant. Think of it as an early study.
4. Materials and Methods: The design of this study seems disorganized and incomplete.
The authors did not make any changes.
Answer: thank you I have corrected and supplemented the points you pointed out.
Reviewer 2 Report
The authors revised the manuscript as recommended by the reviewer. All suggestions have been taken into account and doubtful issues have been carefully clarified by the authors.
Which is why. I believe that the article can be published in its present form.
Author Response
Thank you for the reviewer's compliment and encouragement of our study.
Reviewer 3 Report
The authors have modified the article taking into account all of my previous suggestions
Author Response

(The authors gave the same response as above.)

Round 3
Reviewer 1 Report
Dear authors,
When I read the document, I noticed that most of the recommended changes had been done by the authors. The resubmitted article is now of better quality.